# Comment on Lai et al. Dipeptidyl Peptidase 4 Stimulation Induces Adipogenesis-Related Gene Expression of Adipose Stromal Cells. *Int. J. Mol. Sci.* 2023, *24*, 16101

**DOI:** 10.3390/ijms25137093

**Published:** 2024-06-28

**Authors:** Oscar J Cordero, Martin Kotrulev, Iria Gomez-Touriño

**Affiliations:** 1Department of Biochemistry and Molecular Biology, University of Santiago de Compostela, 15782 Santiago de Compostela, Spain; 2Health Research Institute of Santiago de Compostela (IDIS), 15706 Santiago de Compostela, Spain; m.kotrulev@usc.es (M.K.); iria.gomez.tourino@usc.es (I.G.-T.); 3Centre for Research in Molecular Medicine and Chronic Diseases (CiMUS), University of Santiago de Compostela, 15782 Santiago de Compostela, Spain

**Keywords:** adipogenesis, adiponectin, CCL2, DPP4, PAR-2, stem cells

## Abstract

Adiponectin is a circulating hormone secreted by adipose tissue that exerts, unlike other adipokines such as leptin, anti-inflammatory, anti-atherosclerotic and other protective effects on health. Adiponectin receptor agonists are being tested in clinical trials and are expected to show benefits in many diseases. In a recent article, LW Chen’s group used monocyte chemoattractant protein-1 (MCP-1/CCL2) to improve plasma levels of adiponectin, suggesting the involvement of dipeptidyl peptidase 4 (DPP4/CD26) in the mechanism. Here, we discuss the significance of the role of DPP4, favoring the increase in DPP4-positive interstitial progenitor cells, a finding that fits with the greater stemness and persistence of other DPP4/CD26-positive cells.

The expansion of adipose tissue ascribed to inflammation pathways led by the adipocytes’ hypertrophy (size) is the cause of many health issues, unlike the formation of new adipocytes from precursor differentiation, known as hyperplasia. This process of adipogenesis might have protective effects against metabolic syndrome, atherosclerosis and inflammation. Adiponectin is a circulating hormone secreted by adipose tissue that, unlike other adipokines such as leptin, promotes adipogenesis. Consequently, adiponectin treatment is being tested in small-animal studies as a new therapy that shows promising results in type 2 diabetes mellitus (DM) and obesity [1,2] and has other benefits on health [3].

In a recent *IJMS* article [4], LW Chen’s group proposed dipeptidyl peptidase 4 (DPP4) stimulation as a novel therapy target as it entails local adipogenesis and systemic increases in adiponectin levels. They studied an alternative to not well-established adiponectin receptor agonists, proposing that monocyte chemoattractant protein-1 (MCP-1), known as chemokine (CC-motif) ligand 2 (CCL2), a member of the CC chemokine family, administration improved plasma adiponectin levels in mice, as it enhanced the expression of the DPP4 enzyme.

Although in their manuscript they introduce DPP4, called CD26 in the immune system [5], as an enzyme that cleaves the penultimate L-proline or L-alanine located in the N-terminal region of several, we would like to add some details not covered by the authors.

Several studies assert that DPP4 expression induces a direct proinflammatory effect in several cell types, including macrophages, lymphocytes and smooth muscle cells [6,7]. To be precise, it is soluble DPP4 (sDPP4 or sCD26), secreted or shed from these or other cells [5], acting as a ligand for the protease-activated receptor 2 (PAR-2), a G protein-coupled receptor expressed in many tissues, that causes this effect. The signaling involves the activation of the proinflammatory NFκB pathway [8]. It should be noted that the mechanism could be more complicated if sDPP4 acts cooperatively in complexes, for instance, with coagulation factor Xa, which promotes inflammation and insulin resistance in a rat model of hepatocellular carcinoma (HCC) [9]. In fact, factor X activates the PAR2-RAF1 pathway, synergistically inducing MCP-1 synthesis together with IL-6 in adipose tissue macrophages [10]. In obese mice and humans, the activating transcription factor 4 (ATF4) is upregulated, leading to the enhanced expression of DPP4 and secretion of sDPP4, supporting the outcome obtained in the HCC model, but from another pathway [rev 10]. Of note, more serine-peptidases can signal PAR-2, and there are agonists for this receptor [11].

MCP-1/CCL2, in addition, plays an important role in angiogenesis [12]. Its receptor CCR4, which is also G protein-coupled, is present in many leukocyte subsets [13]. With more DPP4 enzymes, as DPP4 activity cleaves other chemokine ligands of CCR4, such as CCL4 and CCL5 (RANTES), but not CCL2, an imbalance in the migration of—mainly—polymorphonuclear subsets can be expected.

Whether the MCP-1-dependent DPP4 gene upregulation shown by the authors [4] is reflected in the levels of soluble sDPP4 or even in cell-surface PM DPP4 and DPP4 activity is another issue, as DPP4 can be redistributed intracellularly and even exported to the nucleus in a tumor-suppressor p53-dependent manner [10,14,15]. On the other hand, it has been suggested that glypicans (GPC), heparan sulfate structures of the extracellular matrix, behave as co-receptors binding ligands (such as cytokines) and receptors, and GPC3 inhibits DPP4 activity in vitro [16]. Furthermore, there are other biological regulators that are substrates of DPP4, for example, neuropeptides and incretins such as glucagon-like peptide 1 (GLP-1) and glucose-dependent insulinotropic polypeptide (GIP). Due to this fact, there is a family of DPP4 inhibitors, gliptins, that targets type 2 diabetes, a disease closely related to obesity and metabolic syndrome [5,13].

The results shown in the article [4] may have another meaning if we focus on the increase in DPP4-positive interstitial progenitor cells, a subset of preadipocyte PDGFR+ cells that evolve towards intercellular adhesion molecule (ICAM1)-positive but DPP4-negative cells that will generate adipocytes in both subcutaneous and visceral white adipose tissue (WAT) [17]. These adipocyte stem cells previously defined by the expression of PDGFR, a marker of mesenchymal cell populations, fit with the accumulating knowledge acquired in recent years about subsets of cancer stem cells (CSCs) expressing CD26 that have been implicated in metastases of many cancers [18,19,20], including hematological (leukemic stem cells—LSCs, reviewed in [21]). In contrast to other tested antigens co-expressed on chronic myeloid leukemia (CML) LSCs, acute myeloid leukemia (AML) LSCs and normal HSCs, CD26 is the only marker which is not present in CD34+/CD38− normal stem cells. The frequency of these CD26/DPP4+ cells was associated with a poor prognosis in cancer [21] but could be related to greater stemness and persistence, as has been shown in immune system cells [22].

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
