# Peer review of "Comment on Lai et al. Dipeptidyl Peptidase 4 Stimulation Induces Adipogenesis-Related Gene Expression of Adipose Stromal Cells. Int. J. Mol. Sci. 2023, 24, 16101"

_ijms, 2024, doi:10.3390/ijms25137093_

Round 1

Author Response

We would like to thank the reviewer for the constructive comments. We have addressed all Editor's suggestions and questions, and taken the correspondent actions, in the revised manuscript.

Reviewer 2 Report

Comments and Suggestions for Authors

The commentary critically evaluates the methodologies, results, and conclusions of the original study in order to clarify, and expand on, the results presented.

The commentary is well structured and presents a clear and concise critique of the original article "Dipeptidyl Peptidase 4 Stimulation Induces Adipogenesis-Related Gene Expression of Adipose Stromal Cells".

The authors of the comment seems to have a good knowledge of the topic and raises several pertinent points.

Author Response

(The authors gave the same response as above.)

Author Response

(The authors gave the same response as above.)

Round 2

Reviewer 1 Report

Comments and Suggestions for Authors

I have no further comment.

